# Identification of Risk Loci for Radiotoxicity in Prostate Cancer by Comprehensive Genotyping of *TGFB1* and *TGFBR1*

**DOI:** 10.3390/cancers13215585

**Published:** 2021-11-08

**Authors:** Manuel Guhlich, Laura Hubert, Caroline Patricia Nadine Mergler, Margret Rave-Fraenk, Leif Hendrik Dröge, Martin Leu, Heinz Schmidberger, Stefan Rieken, Andrea Hille, Markus Anton Schirmer

**Affiliations:** 1Clinic of Radiotherapy and Radiation Oncology, University Medical Center Göttingen, Robert-Koch-Str. 40, 37075 Göttingen, Germany; manuel.guhlich@med.uni-goettingen.de (M.G.); margret.rf@gmail.com (M.R.-F.); Hendrik.droege@med.uni-goettingen.de (L.H.D.); martin.leu@med.uni-goettingen.de (M.L.); stefan.rieken@med.uni-goettingen.de (S.R.); ahille@med.uni-goettingen.de (A.H.); 2Institute of Clinical Pharmacology, University Medical Center Göttingen, 37075 Göttingen, Germany; laura.hubert@outlook.de (L.H.); Caro.Mglr@gmx.de (C.P.N.M.); 3Department of Radiation Oncology, University Medical Center of the Johannes Gutenberg University, 55131 Mainz, Germany; Heinz.Schmidberger@unimedizin-mainz.de

**Keywords:** radiotherapy, side effects, toxicity, prostate cancer, biomarkers, TGFB, TGBF1, SNP, rs10512263, Leu10Pro, LCL, irradiation

## Abstract

**Simple Summary:**

Genetic variability in transforming growth factor beta pathway (TGFB) has been reported to affect adverse events in radiotherapy. We investigated 40 germline polymorphisms in peripheral blood cells, covering the entire common genetic variability in the TGFβ1 ligand (gene *TGFB1*) and the TGFβ receptor-1 (*TGFBR1*) in 240 patients treated with primary radiotherapy for prostate cancer. Human lymphoblastoid cell lines (LCLs) were used to assess whether *TGFB1* and *TGFBR1* polymorphisms impact DNA repair capacity following single irradiation with 3 Gy. Upon adjustment for multiplicity testing, for one polymorphism (rs10512263 in *TGFBR1*, *C*-variant allele, *n* = 35), a statistically significant association with acute radiation toxicity was observed. As a possible mechanistic explanation, reduced DNA repair capacity in carriers of the *C*-allele after irradiation in LCLs was discovered. This finding has a possible relevance for a plethora of (patho)physiological conditions.

**Abstract:**

Genetic variability in transforming growth factor beta pathway (*TGFB*) was suggested to affect adverse events of radiotherapy. We investigated comprehensive variability in *TGFB1* (gene coding for TGFβ1 ligand) and *TGFBR1* (TGFβ receptor-1) in relation to radiotoxicity. Prostate cancer patients treated with primary radiotherapy (*n* = 240) were surveyed for acute and late toxicity. Germline polymorphisms (*n* = 40) selected to cover the common genetic variability in *TGFB1* and *TGFBR1* were analyzed in peripheral blood cells. Human lymphoblastoid cell lines (LCLs) were used to evaluate a possible impact of *TGFB1* and *TGFBR1* genetic polymorphisms to DNA repair capacity following single irradiation with 3 Gy. Upon adjustment for multiplicity testing, rs10512263 in *TGFBR1* showed a statistically significant association with acute radiation toxicity. Carriers of the *Cytosine (C)*-variant allele (*n* = 35) featured a risk ratio of 2.17 (95%-CI 1.41–3.31) for acute toxicity ≥ °2 compared to *Thymine/Thymine (TT)-*wild type individuals (*n* = 205). Reduced DNA repair capacity in the presence of the *C*-allele of rs10512263 might be a mechanistic explanation as demonstrated in LCLs following irradiation. The risk for late radiotoxicity was increased by carrying at least two risk genotypes at three polymorphic sites, including Leu10Pro in *TGFB1*. Via comprehensive genotyping of *TGFB1* and *TGFBR1*, promising biomarkers for radiotoxicity in prostate cancer were identified.

## 1. Introduction

Prostate cancer is the most frequent malignancy in male patients and accounts for about 20% of new cases per year [1]. Apart from an active surveillance strategy in localized tumors with favorable prognostic parameters, radical prostatectomy and radiation therapy are the two options when curation is intended [2]. Recent analyses based on nation-wide datasets in Sweden demonstrate, upon adjustment, similar prostate cancer mortalities for these two regimens [3]. Given comparable tumor control rates, therapy-related risks should be considered in particular when informing patients and deciding on a specific treatment. Despite appreciable advances in surgical urology, there still remain considerable risks of radical prostatectomy and subsequent quality of life-impairing inconveniences, mainly concerning urinary and sexual function [4]. Acute and late side effects of radiotherapy usually affect the rectum and the urinary bladder [5]. Such events occurring up to three months upon therapy completion are defined as acute, thereafter as late toxicity. There seems to be a link between acute and late toxicities in radiotherapy of prostate cancer. Patients with relevant acute symptoms experienced a 42% risk for late sequelae of least grade 2 in contrast to a risk of only 9% for those without major acute reactions [6]. Various attempts have been undertaken to reduce the risk of toxicity, e.g., by applying spacers between the prostate and rectum [7]. Alongside non-genetic factors, such as patient age, radiation technique, dosage, and the extent of the irradiated field, the intrinsic radiosensitivity driven by the genetic make-up is thought to be a major contributor for radiation toxicity [8,9]. Integration of these different kinds of information seems to enhance prediction of adverse radiation effects [10]. With better control of the extrinsic factors, the focus shifts to genetics exploiting the enormous technical advances in this field. Identification of genetic variation linked to adverse effects of irradiation may allow to define individuals at risk prior to therapy [11].

The transforming growth factor beta (TGFβ) pathway has gained major attention in the field of radiotherapy, promoting both radioresistance of malignant cells and injury of normal tissues [12]. Radiation is one of the key activators of the TGFβ1 ligand, releasing the biologically active from its latent form [13]. Moreover, radiation was reported to increase TGFβ1 expression in a rat liver model in a dose-dependent manner over several months [14]. A pre-treatment elevated TGFβ1 was linked to worsened echocardiography upon adjuvant radiotherapy for breast cancer [15]. Genetic polymorphisms might modulate the expression of TGFβ1 [16]. Interestingly, the Leu10Pro substitution in the signal peptide of TGFβ1 was suggested as a biomarker for the risk of radiation-induced pneumonitis, the most threatened side effect of radiotherapy [17]. Whilst most reports addressing the impact of TGFβ pathway polymorphisms on side effects of radiotherapy so far are retrospective, a recent study conducted in a prospective fashion highlighted the −*509T* allele in the promoter region of *TGFB1* as a key determinant of breast fibrosis risk [18]. With regard to prostate cancer, only a few retrospective reports have addressed a potential impact of TGFβ pathway genetics in relation to radiation-induced toxicity, leaving this issue still under debate [19,20,21].

With the availability of a lot of data on human genetic variation, maps of linkage disequilibrium between genetic markers can be constructed [22,23]. This data enables researchers to define sets of markers that comprehensively cover the common genetic variability in a region of interest. We applied this approach on the genetic regions of two human genes: *TGFB1* (encoding the TGFβ1 ligand) and *TGFBR1* (TGFβ receptor 1). We demonstrate for the first time, to the best of our knowledge, a strong association of a single genetic marker in *TGFBR1* with acute toxicity and provide functional data for putative mechanistic actions. In addition, by a panel of three markers, a good prediction of late toxicity was possible, with one of these markers in the literature repeatedly attributed to the risk for late sequelae of radiotherapy applied to other malignant entities.

## 2. Materials and Methods

### 2.1. Patient Cohort

The study is based on a cohort of 509 patients with prostate cancer treated consecutively at a single institution (Department of Radiotherapy and Radiooncology, Göttingen, Germany) between 2001 and 2011. Patients were recruited in accordance with the ethical standards of the responsible institutional committee on human experimentation and with the Helsinki Declaration of 1975, as revised in 2000. The study was approved by the local university ethical committee (application numbers 6/6/96 and 22/9/04). Written informed consent was obtained by all study patients. All patients were diagnosed with biopsy-proven adenocarcinoma of the prostate gland and were staged according to the respective UICC criteria and the Gleason score.

For evaluation of genetic markers with clinical toxicity, the only patients considered were those who received primary external beam radiotherapy with or without high-dose-rate (HDR) brachytherapy, finally resulting in 240 eligible individuals (see flowchart Figure 1).

### 2.2. Administration of Radiotherapy

Details of the radiotherapy and toxicity scoring have been previously described [24,25]. Briefly, patients received definitive therapy to the prostate bed with radiation doses between 64–77 Gy (median dose 72 Gy) and doses and planning target volumes (PTVs) were adjusted to disease stage and risk factors. Only two patients got more than 72 Gy. In one patient, 77 Gy was prescribed upon treating physician’s discretion and patient’s choice. In another patient, radiotherapy was interrupted due to acute epididymitis and was completed to a final dose of 74 Gy.

The clinical target volume (CTV) for the treatment of the prostate was specified as including the prostate and, if indicated, the proximal or whole seminal vesicles, plus 1 cm surrounding margin, resulting in the PTV. The irradiation technique included individual optimization with conformal treatment planning, the use of multiple radiation fields, individual blocks, rectal balloons, or the prone position with a belly board, if possible, to reduce the small bowel volume within the PTV.

Depending on risk factors, patients were irradiated to the pelvic lymph nodes to 45 Gy. Selected patients with prostate carcinoma stage T2b-T3a, T1-T2a with a Gleason-score ≥ 7, and/or PSA ≥ 10 underwent a combined external beam radiotherapy and HDR brachytherapy. The latter was done with Swift^®^ (Nucletron Corporation B.V., Veenendaal, The Netherlands). Patients received two fractions with 9 Gy at an interval of 2 weeks.

### 2.3. Scoring of Radiation Toxicity

Acute and late genitourinary and gastrointestinal side effects were documented according to the Common Toxicity Criteria Adverse Events (CTCAE) definition, version 3.0, or the LENT SOMA scale, version 3.0, respectively. Acute toxicity was monitored weekly during RT, and late toxicity was recorded during follow-up. For analyses, toxicity in terms of proctitis and cystitis was considered. Organ toxicity ≥ grade 2 in at least one organ system was chosen as the cut-off value, as in patients with toxicity ≥ grade 2, quality of life is significantly impaired. The investigators responsible for scoring patients’ toxicity and performing genotype analyses were blinded to each other’s results.

### 2.4. Selection and Typing of Genetic Polymorphisms

The common genetic diversity, i.e. at least 5% minor allele frequency (MAF), of the *TGFB1* and *TGFBR1* regions was analyzed in Caucasians based on the linkage disequilibrium data from the 1000 human genome database (http://www.internationalgenome.org, accessed on 19 February 2018). For visualization and marker tagging, HaploView software was used (www.broadinstitute.org/haploview/haploview, accessed on 25 February 2018 [26]). The respective graphs are shown as Appendix A (for *TGFB1*) and Appendix A (*TGFBR1*) in the Appendix A. Thereof, 40 marker sites, 27 in *TGFB1* (Appendix A) and 13 in *TGFBR1* (Appendix A), were selected to cover comprehensively the panel of genetic polymorphisms with a MAF of ≥5%. These sites were genotyped by the primer extension method in DNA isolated from peripheral blood cells of the patients (for primer sequences see Appendix A). For details on polymorphism selection and technical issues of genotyping, please refer to the online Appendix A.

### 2.5. Assessment of DNA Repair Capacity

A panel of 189 Epstein Barr Virus-immortalized lymphoblastoid cell lines (LCLs) of Caucasian origin obtained from the Coriell Institute (https://www.coriell.org/, accessed on 25 January 2020) was employed as a model to investigate DNA damage repair upon genotoxic exposure. The applied procedure was based on staining of residual gamma-H2AX (γH2AX) foci according to a published protocol [27]. Cells were cultured in RPMI-1640 medium, supplemented with 15% fetal calf serum and 1% penicilline/streptomycine (providers for all chemicals, kits, buffers, and cell culture medium are provided in Appendix A). LCLs were counted by flow cytometry on a FACScan™ machine (Becton Dickinson, in the following abbreviated as “BD”, Franklin Lakes, NJ, USA) using CountBright™ beads (Thermo Fisher Scientific, Waltham, MA, USA). Only LCL cultures with at least 50% viable cells, as determined by propidium iodid (BD) staining in flow cytometry using CellQuest™ software (BD) for analysis, were used for subsequent experiments. Upon counting and centrifugation (at 250 g, 5 min, room temperature), cells were transferred to 6-well plates (Nunclon™ Delta Surface, Thermo Fisher Scientific), with 2.5 mL per well at a density of 200,000/mL, in duplicates for each experimental condition. Cells were exposed to 5-fluorouracil (5-FU) at a final concentration of 3 µM, intended as a radiosensitizer. Parallel experiments were conducted with or without human recombinant TGFβ1 at 5 ng/mL in order to compare effects with specific TGFβ pathway stimulation. For each LCL, samples with 5-FU ± TGFβ1 and a control with cell culture medium only were incubated for 16 h at 37 °C with 5% CO_2_.

Thereafter, the cells were irradiated once at 3 Gy using RS225 X-ray device (Gulmay Medical, Byfleet, UK) with the specified settings: 1 Gy/min, 200 kV, 15 mA, 0.5 mm cupper filter, 500 mm table height. The single fraction of 3 Gy was chosen according to prior dose-response assessments. This dose caused significant amounts of γH2AX foci without resulting in overt cytotoxicity in order to maintain processes allowing DNA repair. Upon irradiation, cells were further incubated at 37 °C and 5% CO_2_ for 6 h. Then, 1 mL of each cell sample was pipetted into 5 mL tubes and placed on ice, with all following steps carried out at 4 °C. Cells were washed in ice-cold PBS and resuspended in 50 µL of resuspension buffer (for details on buffer compositions please refer to Appendix A). FACS tubes were prepared with 150 µL of a solution of Block-9 staining buffer, supplemented with 0.6 µg/mL of the FITC-conjugated anti-phospho-histone H2A.X (Ser139) antibody and clone JBW301, which binds to residual γH2AX foci. Of the resuspended cells, 20 µL were transferred to the prefilled FACS tubes and stained for 3 h at 4 °C protected from light. Subsequently, 350 µL of 1:2,000-diluted Vibrant^®^ DyeCycle™ (Thermo Fisher Scientific) was added to the cell suspension in the FACS tubes for 30 min at 37 °C. Assessment was then carried out via fluorescence-activated cell sorting using a LSR-II device with B-Diva software (BD). Signals of γH2AX foci staining (assessed in the FITC channel) were first referred to the DNA content (APC channel). The obtained ratios for the specifically treated samples (5-FU + 3 Gy irradiation with and without TGFβ1) were then related to a control sample of the respective LCL solely treated with cell culture medium. These ratios represent the final read-out parameter for association testing with genotypes.

## 3. Results

### 3.1. Toxicity Distribution

Data concerning acute toxicity were available for all eligible 240 patients, and late toxicity data by 10/2019 were recorded for 238 patients (two did not keep to the aftercare dates). Combined toxicity was considered the primary outcome parameter defined as the maximum grade of the single items observed at the rectal and urinary bladder, each for acute and late toxicity. Overall, 30.0% and 19.3% of the patients developed acute and late side effects ≥ grade 2, respectively (Table 1). No acute toxicity was documented in 30 patients (12.5%), mild as grade 1 in 139 patients (57.9%), intermediate as grade 2 in 69 patients (28.8%), and severe as grade 3 in 2 patients (0.8%). Late radiation toxicity was absent in the majority of patients (54.1%), mild in 25.8% of patients, intermediate in 15.8% of patients, and severe in 3.3% of patients (see Appendix A online, including data for single organ toxicities). No life-threatening or lethal toxicities (grades 4 and 5) occurred, neither acute nor late.

### 3.2. Genotyping Performance

In total, 9600 genetic positions were investigated (240 samples at 40 loci). Only 19 positions were finally left undetermined (no clear signals in the electropherogram and/or equivocal results upon repetitions), yielding an overall mean call rate of 99.8%. Repetition at 2522 positions confirmed genotyping accuracy. Genotyping was in accordance with the Hardy-Weinberg equilibrium for all 40 polymorphisms (all *p* > 0.1, for a complete list and allelic frequencies please see Excel sheet “Appendix A”). LD of the 27 *TGFB1* and the 13 *TGFBR1* loci analyzed in the 240 prostate cancer patients is illustrated in Appendix A. These analyses revealed r² values > 0.80 for some pairwise comparisons in this dataset, i.e., there was some redundancy in genotyping.

### 3.3. Genetic Polymorphisms and Radiation Toxicity

The 40 assayed genetic polymorphic sites were first assessed in a univariable fashion for associations with acute and late radiation toxicity. Genotypic configurations, with at least 10 patients, were considered for statistical analyses. Combined toxicities of the rectum and urinary bladder served as a primary endpoint, while those of both organs served separately as secondary endpoints. The full data set of these genetic analyses is provided in the Excel sheet “Appendix A”. Table 2 depicts univariable nominal associations at *p* < 0.1 for patient and treatment-related parameters., as well as for genetic polymorphisms (assuming either a dominant or recessive variant allele effect) in relation to combined acute or late radiation toxicities. Patients who experienced acute toxicity ≥ 2 had a 2-fold higher risk for sequential late toxicity ≥ 2 (*p* = 0.002). Not reaching statistical significance, inclusion of the pelvic lymphatic drainage area into the radiation volume showed a trend towards increased acute toxicity. Four genetic parameters were associated at *p* < 0.1 with acute toxicity in univariable analysis: rs1800470 and rs2241713 in *TGFB1*; rs10512263 and rs34733091 in *TGFBR1.* Analogously, late toxicity was impacted by six genetic markers: rs10417924, rs1800470, rs2241713, rs75041078, rs8108357 in *TGFB1*, and rs78471739 in *TGFBR1*. When a statistically significant pairwise LD was defined at *p* < 0.001 (conservative approach), only 10 out of the 40 genetic polymorphisms tested were considered as independent variables. With testing combined acute and late toxicity as primary endpoints, the adjusted threshold of statistical significance was thus set at *p* = 0.05/10/2 = 0.0025. In univariable analysis, this cut-off was reached only for *TGFBR1* rs10512263 in regard to acute toxicity, with a risk ratio of 2.14 (95%-CI: 1.46–3.15) for carriers of at least one variant *C*-allele at this site.

Next, based on univariable associations of *p* < 0.1 (Table 2), we set up multivariable analyses combining genetic and non-genetic markers. This was executed by stepwise forward conditional binary logistic regression models. Radiation of the pelvic lymphatic drainage area and two genetic polymorphisms were associated at *p* < 0.1 with acute toxicity (Table 3). This analysis confirmed a strongly increased risk for acute toxicity ≥ °2 for the minor allele at rs10512263 and a weaker protective effect of the proline allele at rs1800470. In regard to late toxicity ≥ °2, this multivariable analysis highlighted acute toxicity ≥ °2 (OR 2.62, 95%-CI: 1.31–5.21, *p* = 0.006) and the homozygous variant allele (corresponding to proline in the encoded protein) at *TGFB1* rs1800470 (2.70, 1.11–6.53, *p* = 0.028) as risk factors, whereas the variant alleles at *TGFB1* rs10417924 (0.39, 0.18–0.86, *p* = 0.019) and at *TGFBR1* rs78471739 (0.16, 0.02–1.20, *p* = 0.075) turned out as protective.

Additive effects of the number of risk genotypes at these three genetic loci were observed (Figure 2). In comparison to patients with none of these risk genotypes, carriers with two or three of them exhibited a RR of 2.94 (95%-CI: 1.44–6.02, *p* = 0.001, according to Fisher’s two-sided exact test) to encounter late radiation sequelae of at least grade 2, which is certainly relevant in terms of quality of life.

### 3.4. Impact of rs10512263 on DNA Repair

The TGFβ pathway was reported to protect cells from radiation damage [29]. We sought to investigate whether the considered polymorphisms in *TGFB1* or *TGFBR1* genetic region may affect DNA repair upon exposure to a single irradiation of 3 Gy in combination with 5-FU as radiosensitizer with or without additional TGFβ1 ligand added for TGFβ pathway stimulation. As a model, lymphoblastoid cell lines were used. To enhance specificity for TGFβ signaling, only associations at *p* < 0.05 upon stimulation with TGFβ1 ligand (in combination with irradiation and 5-FU as radiosensitizer) were considered relevant. While none of the *TGFB1* polymorphisms (Figure 3a, statistical evaluation by Jonckeere-Terpstra trend test taking into account allele dosage effects) matched this criterion, two of *TGFBR1* did (Figure 3b). The strongest signal for modifying DNA repair (measured as residual γH2AX foci) was found for the *TGFBR1* polymorphism rs10512263. Cells harboring the minor *C*-allele variant (26 heterozygous and one homozygous cell lines) exhibited a higher rate of residual γH2AX foci compared to the 163 LCLs homozygous for the wild type *T*-allele (*p* = 0.005, Mann-Whitney U test, Figure 3c). Without the addition of TGFβ1, this association was weaker, albeit still present (*p* = 0.047). The second polymorphism associated with DNA residual γH2AX foci at *p* < 0.05 was *TGFBR1* rs34733091 (Figure 3d); however, it was not independent from rs10512263 as there is substantial genetic LD between these two markers (see Appendix A).

## 4. Discussion

We applied comprehensive genotyping for *TGFB1* and *TGFBR1* polymorphisms in relation to acute and late side effects of radiotherapy in prostate cancer. Concerning acute radiation toxicity, a statistically significant association upon multiple testing adjustment was observed for the SNP rs10512263 located in intron 1 of *TGFBR1*. In a multivariable logistic regression model, this SNP conferred a remarkably high-risk ratio of 3.80 (95%-CI 1.78–8.12, *p* = 0.0006) for encountering acute toxicity of at least °2 in irradiation of primary prostate cancer. In addition, the variant allele of this SNP, which was associated with an increased risk for acute radiation toxicity, showed the strongest signal for impaired DNA repair upon irradiation in 189 cell lines. Regarding late radiation toxicity, we could not demonstrate a statistically significant association for any of the assayed markers upon adjustment for multiple testing. However, a combination of three genetic markers rather than a single one elicited a substantial risk condition.

There is at present only sparse information on *TGFBR1* rs10512263 in literature, and the respective data are inconsistent. Regarding the minor variant allele of this SNP, for which our study indicates a reduced DNA repair capacity and an increased risk for acute radiation toxicity, a first report attributed a protective effect in regard to breast cancer susceptibility [30]. However, two subsequent studies found an increased risk along with this allele for gastric and endometrial cancer [31,32].

Several mechanisms have been elucidated as to how DNA integrity might be preserved by TGFβ1. It fosters error-free homologous-recombination of DNA double-strand-break, thereby enhancing resistance of cells towards genotoxic stressors, such as radiotherapy [33]. Moreover, non-homologous end-joining is facilitated by TGFβ1, also counteracting DNA damage induced by irradiation [34]. In contrast, loss of TGFβ1 was reported to shift cells to an alternative, more error-prone “backup” pathway of DNA repair, rendering cells more sensitive to genotoxic therapies [35]. Possibly, the higher rate of residual γH2AX foci upon irradiation seen in our study in the presence of the variant allele of the *TGFB1* rs10512263 polymorphism may be due to a mitigated TGFβ1 activity, resulting in less effective DNA repair. Given the link to DNA integrity, this could have implications in carcinogenesis. Inhibition of TGFβ1 has been reported to increase radiosensitivity via reduced DNA damage repair including blocked γH2AX foci formation [29]. Conversely, over-expression or over-activity of TGFβ signaling is a frequently observed feature in progressed malignant diseases, possibly explaining equivocal associations of TGFβ pathway genetic polymorphisms with cancer traits, as stimulation of TGFβ signaling immediately following irradiation was linked to anti-inflammatory conditions in a cellular model [36,37]. Thus, a less active genetic variant as the putative *C*-allele of *TGFBR1* rs10512263 may increase the clinical toxicity risk.

One of the three markers related to late radiation toxicity comprises *TGFB1* rs1800470, i.e., the Leu10Pro substitution in the signal peptide of the respective protein. This polymorphism has frequently been assessed regarding side effects of radiotherapy. An extensive study launched to test formerly reported associations of candidate gene markers (including *TGFB1*, but not *TGFBR1*) for impact on early or late side effects of radiotherapy in breast and prostate cancer did not prove a statistically significant risk for any of the markers tested [38]. However, in that study, associations were evaluated on a per-genotype basis, i.e., assuming a co-dominant effect model. Similarly, we also could not find late radiation toxicity related with rs1800470 (*p* > 0.2). Only when considering a recessive effect model we could delineate the impact of this marker. Furthermore, genotyping of rs1800470 is challenging due to an extraordinarily high *CG* content in this region. Using the Hardy-Weinberg equilibrium (HWE) as quality criteria for genotyping, we could achieve an almost perfect result (*p* = 0.95 for deviation from HWE). However, though the respective HWE in the referenced study (*p* = 0.08) appears acceptable, the chance for miss-classification of genotypes rises with decreasing *p* values [38]. In that study, there was a large variation of total radiation dosage (57–74 Gy), and numbers of fractions (*n* = 19–37) varied largely in the subset of patients recruited with prostate cancer. It is particularly important that the *T*-allele of the SNP rs1800469, which is high in LD with the *C*-allele of rs1800470, was identified as a risk condition for chronic tissue fibrosis upon breast irradiation in the first prospectively randomized genomic marker validation study [18]. Moreover, prostate cancer patients carrying the homozygous genotype of this variant allele experienced an enhanced risk for late rectal bleeding upon radiotherapy [21]. This is noteworthy since the corresponding genotype at rs1800470 was linked to late toxicity in our study. Mechanistically, a markedly increased secretion of TGFβ1 in a cellular model system and elevated TGFβ1 serum levels were noticed for the proline compared to the leucine allele of this SNP [39,40].

The second *TGFB1* marker related to late radiation toxicity, rs10417924, was not covered by the marker panel assayed in the aforementioned study [21]. There is not much known about this SNP in literature apart from a suggested relationship with the risk for B-cell acute lymphoblastic leukemia (B-ALL) in children [41]. The variant allele, which was less frequent in B-ALL cases than in controls, evolved as protective regarding late radiation toxicity in our study. For the third marker (*TGFBR1* rs78471739), for which we observed an increased risk for late radiation sequelae, there is at present no reference in the literature.

We want to point out that we, in line with contemporary literature consensus, could not identify a single marker with statistically significant association with late radiation toxicity upon prostate cancer irradiation when considering multiple testing. Heterogeneity in radiation sites, dosages, and fractionations are likely to impair association analyses between genotypes and side effects, resulting in the absence of statistically significant relationships in a meta-analysis focused on the three most frequently assessed *TGFB1* SNPs [42].

Furthermore, and in line with literature, we confirmed acute radiation toxicity as a relevant factor for the risk to develop late sequelae [6]. In this regard, technical advances may be beneficial as intensity-modulated radiation therapy (IMRT) combined with image-guided radiation therapy (IGRT) was reported to decrease acute and late rectal toxicity following prostate cancer radiotherapy [43].

This study is limited by its retrospective nature. Although we extracted eligible patients starting from a panel of 509 to end up with 240 individuals, there is still some degree of heterogeneity in this sample set. This concerns, *inter alia*, the total radiation doses in a range from 64 to 77 Gy and HDR, which 26.7% of our patients received in combination with external radiotherapy. Even if a univariable analysis (Table 1) did not elicit a statistically significant impact of either total radiation dose or HDR, we cannot completely exclude bias on acute or late toxicity. Another limitation is the use of LCLs instead of prostate cancer cells for mechanistic studies. However, unlike prostate cancer cells, the LCL panel enabled us to study DNA repair assessment dependent on genotypes in a large number of cell lines (*n* = 189), comparable to the eligible prostate cancer cohort (*n* = 240). LCLs and prostate cancer cells are certainly distinct, but we think that early radiation responses by formation and repair of γH2AX foci are conserved and shared by different cell types. In lymphocytes exposed to a single high dose irradiation, higher decay ratios of γH2AX foci reflecting a more proficient DNA repair capacity was correlated with less normal tissue toxicity of irradiated patients [44].

The data reported here may have important clinical relevance. If confirmed in follow-up studies, at best in prospective fashions, pre-therapeutic assessment of relevant biomarkers may assist in treatment tailoring. This may be of particular interest, if equivalent strategies are available, i.e., radiotherapy or surgery for primary treatment of prostate cancer. In addition, the biomarkers identified here may be applicable to other entities treated with radiotherapy as well. Finally, our results may contribute to a better understanding of the complex mechanisms linking TGFβ signaling with DNA repair and radiosensitivity. Further investigations might be stimulated by our findings.

## 5. Conclusions

We identified the single nucleotide polymorphism rs10512263 in *TGFBR1*, robustly linked it to acute radiation toxicity, and provided a possible mechanistic rationale for this association. Thus, we propose rs10512263 as a putative biomarker to be further studied in relation to acute side effects of radiotherapy in prostate and possibly other cancer entities. In regard to long-term radiotoxicity, our data suggest a possible relevance for a combination of three markers, among them is the frequently studied Leu10Pro polymorphism. Additional clinical and functional research should further delineate the medical and biological significance of the findings reported here, having the potential to impact treatment decisions for patients.

## Figures and Tables

**Figure 1 cancers-13-05585-f001:**
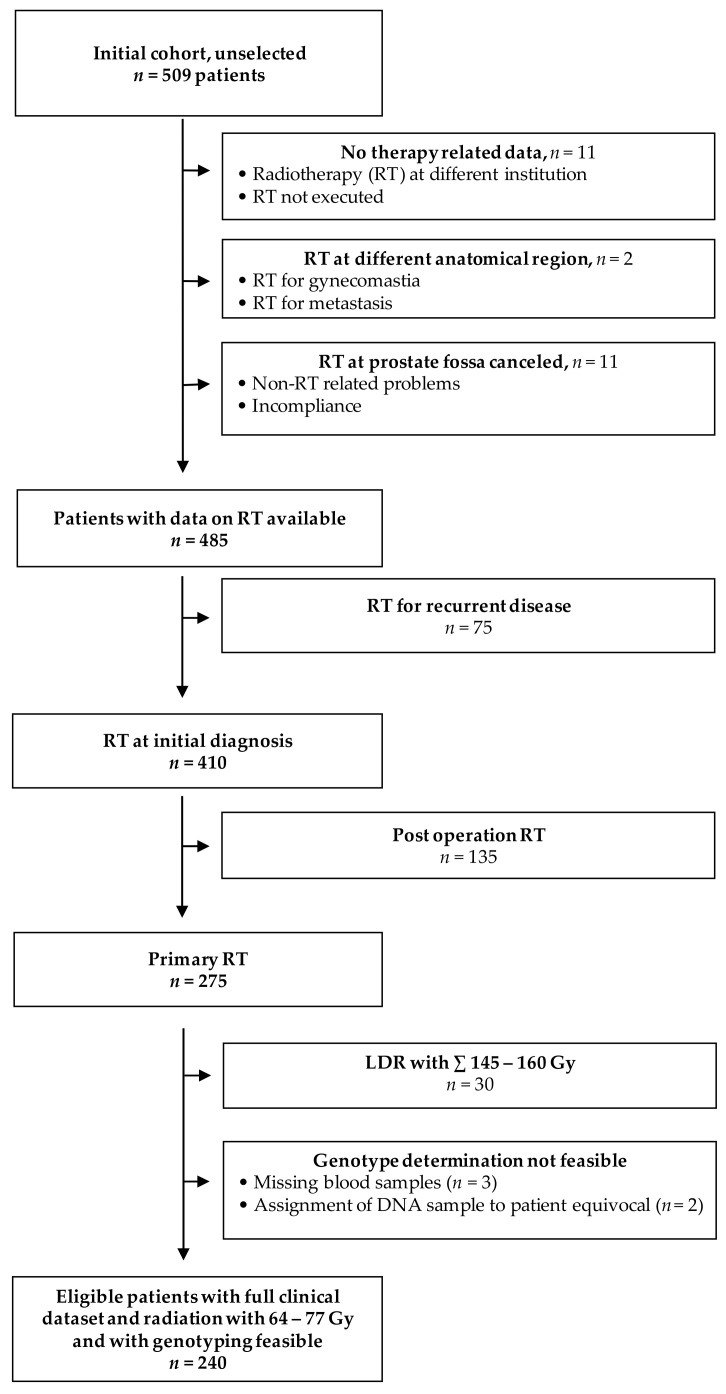
Flowchart of patient selection for study cohort. LDR = low dose rate radiotherapy (i.e., brachytherapy with permanent seeds).

**Figure 2 cancers-13-05585-f002:**
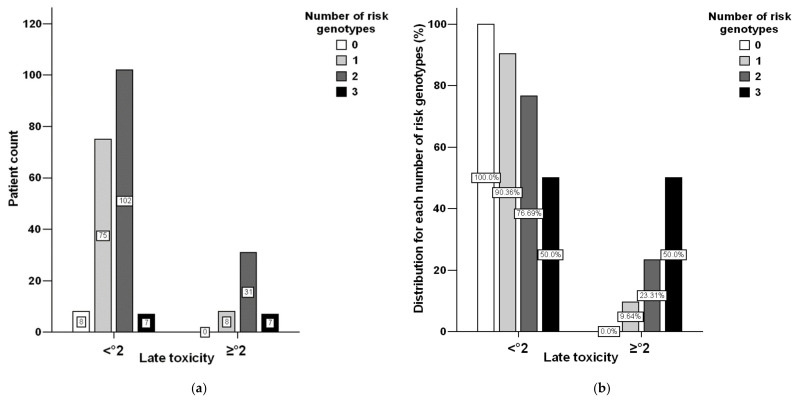
Late radiation toxicity stratified by none or mild (<°2) versus medium or high sequelae independence on the numbers of three risk genotypes (*Guanine*/*Guanine* at *TGFB1* rs10417924, *Cytosine*/*Cytosine* at *TGFB1* rs1800470, and *Thymine*/*Thymine* at *TGFBR1* rs78471739). (**a**) Numbers of patients, (**b**), relative distribution of risk genotypes.

**Figure 3 cancers-13-05585-f003:**
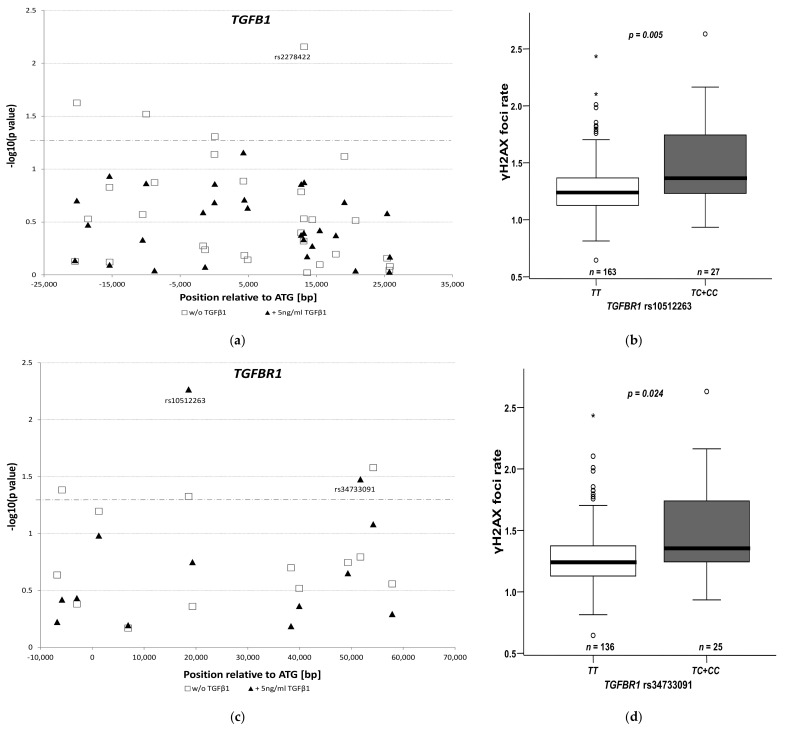
*TGFB1* (panel (**a**)) and *TGFBR1* (panel (**b**)) genetic polymorphisms in relation to residual γH2AX foci. Upon single irradiation of 3 Gy (with 3 µM 5-fluorouracil used as radiosensitizer), with (filled triangles) or without (open squares) 5 ng/mL TGFβ1, LCLs were subsequently incubated for 6 h at 37 °C to allow for DNA repair prior to γH2AX staining. Data were each normalized to total DNA content (by co-staining with allophycocyanin) and to only medium-treated control cells. (**c**,**d**) Visual depiction of residual γH2AX foci rate with TGFβ1 stimulation dependent on genotypic configurations at *TGFBR1* rs10512263 and rs34733091. The boxplots indicate the data distributions as follows: The rectangle represents 50% of the values of a given distribution with the lower horizontal line reflecting the 25%- (Q_25_), and the upper the 75%-(Q_75_) quartile. The difference of these two delimiters is called the interquartile distance (IQA). Values within 1.5-times of the IQA below Q_25_ or 1.5-times above Q_75_ are depicted by the whiskers of the blot (vertical line limited by short horizontal). Values out of this range are either marked by circles (>1.5-times, but ≤ 3-times of IQA referred to Q_25_ or Q_75_) or asterisks (beyond 3-times of IQA on either side). Abbreviations: *ATG* = initiation codon; bp = base pairs, w/o = without; *TT = Thymine/Thymine*; *TC = Thymine/Cytosine*; *CC = Cytosine/Cytosine*.

**Table 1 cancers-13-05585-t001:** Patient baseline, disease, and treatment data.

Parameter	All	Acute Toxicity ≥ °2 *n* = 71	Late Toxicity ≥ °2 *n* = 46
Age (years), median (IQR, min-max)	70 (67–73, 53–83)	69 (66–74, 57–79)	70 (67–74, 62–79)
		*p* = 0.688	*p* = 0.178
BMI (kg/m²), median (IQR, min-max)	27 (25–30, 19–44)	27 (25–30, 21–41)	27 (25–28, 22–41)
		*p* = 0.708	*p* = 0.329
T stage, No. (%)			
1	60 (25.0)	17 (23.9)	14 (30.4)
2	155 (64.5)	45 (63.4)	26 (56.5)
3	21 (8.8)	8 (11.3)	5 (10.9)
4	4 (1.7)	1 (1.4)	1 (2.2)
		*p* = 0.622	*p* = 0.829
N stage, No. (%)			
cN0	234 (97.5)	67 (94.4)	45 (97.8)
cN1 $	6 (2.5)	4 (5.6)	1 (2.2)
		f.s.	f.s.
Gleason score #, No. (%)			
≤5 &	46	10	9
6	104	34	18
7	68	22	14
≥8	18	4	5
		*p* = 0.538	*p* = 0.511
PSA at diagnosis (ng/mL), median (IQR, min-max)	9.4 (6.2–15.6, 1.1–186)	10.0 (7.1–19.9, 2.1–179)	8.6 (6.2–13.9, 1.7–77)
		*p* = 0.174	*p* = 0.510
PLDA, No. (%)			
No	190 (79.2)	51 (71.8)	39 (84.8)
Yes	50 (20.8)	20 (28.2)	7 (15.2)
		*p* = 0.063	*p* = 0.280
HDR, No. (%)			
No	176 (73.3)	55 (77.5)	37 (80.4)
Yes	64 (26.7)	16 (22.5)	9 (19.6)
		*p* = 0.600	*p* = 0.319
RT dose, median	72	72	72
(IQR, min-max)	(68.4–72, 64–77)	(71–72, 64–72)	(71–72, 66–72)
		*p* = 0.526	*p* = 0.223
ADT, ever, No. (%)			
No	77 (32.1)	19 (26.8)	14 (30.4)
Yes	163 (67.9)	52 (73.2)	32 (69.6)
		*p* = 0.253	*p* = 0.861
ADT, concomitant to RT §, No. (%)			
No	84 (35.0)	20 (28.2)	14 (30.4)
Yes	156 (65.0)	51 (71.8)	32 (69.6)
		*p* = 0.152	*p* = 0.524
Follow-up ‡, median (IQR, min-max)	59.5 (49.8–62.5, 0.0–150.6)	55.7 (47.5–61.8, 0.6–130.5)	54.0 (40.5–62.2, 12.7–109.3)

Abbreviations: ° = grade (in toxicity scoring), IQR = interquartile range; BMI = body mass index; EB = external beam; HDR = high dose rate; PLDA = pelvic lymphatic drainage area, RT = radiotherapy, ADT = androgen deprivation therapy, f.s. = few samples. *p* values indicate comparisons between each acute and late toxicity ≥ °2 versus < °2, as assessed by binary logistic regression. $ As only six patients were classified as cN1, this number was considered too little for statistical analysis. # In four patients, of whom one with acute toxicity ≥ °2, the Gleason score could not be determined due to insufficient tumor material available. & Combined Gleason scores <6 only concerns patients recruited before 2005 when the International Society of Urological Pathology decided not to assign scores <6 to “carcinoma” anymore [28]. § ADT administered parallel to RT or stopped no longer than 3 months before. ‡ Follow-up was counted from the last day of radiation.

**Table 2 cancers-13-05585-t002:** Univariable analysis for association of non-genetic and genetic parameters with combined acute (according to CTCAE) and late (LENT-SOMA) proctitis or cystitis of grade ≥ 2.

Parameter	RR (95%-CI), Puni, Raw *	
	Acute	Late
Non-genetic		
Radiation field: with PLDA vs. without	1.51 (1.00–2.29), 0.080	
Acute toxicity ≥ °2	-	2.00 (1.37–2.92), 0.002
Genetic		
*TGFB1* rs1800470 (L10P): *TC* + *CC* vs. *TT*	0.71 (0.48–1.05), 0.087	
*TGFB1* rs2241713: *GC + CC* vs. *GG*	0.70 (0.47–1.03), 0.084	
*TGFBR1* rs10512263: *TC + CC* vs. *TT*	2.14 (1.46–3.14), 0.001	
*TGFBR1* rs34733091: *TC + CC* vs. *TT*	1.61 (1.07–2.42), 0.034	
*TGFB1* rs10417924: *GA + AA* vs. *GG*		0.52 (0.28–0.97), 0.041
*TGFB1* rs1800470 (L10P): *CC* vs. *TT* + *TC*		1.95 (1.11–3.45), 0.035
*TGFB1* rs2241713: *CC* vs. *GG + GC*		1.95 (1.11–3.45), 0.035
*TGFB1* rs75041078: *TT* vs. *CC + CT*		2.17 (1.14–4.14), 0.056
*TGFB1* rs8108357: *GG* vs. *AA + AG*		2.02 (1.15–3.56), 0.029
*TGFBR1* rs78471739: *TC + CC* vs *TT*		0.17 (0.02–1.16), 0.022

Abbreviations: CTCAE = Common Toxicity Criteria assessment for Adverse Events; LENT-SOMA = Late Effects of Normal Tissues − Subjective, Objective, Management criteria with Analytic laboratory and imaging procedures; RR = relative risk ratio; CI = confidence interval; TC = Thymine/Cytosine; CC = Cytosine/Cytosine; TT = Thymine/Thymine; CC = Cytosine/Cytosine; GG = Guanine/Guanine; GA = Guanine/Adenine; AA = Adenine/Adenine; PLDA = pelvic lymphatic drainage area; *TGFB1* = transforming growth factor beta-1; *TGFBR1* = transforming growth factor beta receptor-1; rs<number> = unique reference number for single nucleotide polymorphisms (SNPs) according to the dbSNP database (www.ncbi.nlm.nih.gov/projects/SNP/, accessed on 19 February 2018). * Fisher’s exact test.

**Table 3 cancers-13-05585-t003:** Multivariable analysis for association of non-genetic and genetic parameters with combined acute (according to CTCAE) or late (LENT-SOMA) proctitis or cystitis of grade ≥ 2.

Parameter	OR (95%-CI), Puni, Raw *	
	Acute toxicity	
Radiation field: with PLDA vs. without	1.87 (0.95–3.69), 0.072	
*TGFB1* rs1800470 (L10P) #: *TC + CC* vs. *TT*	0.53 (0.29–0.95), 0.034	
*TGFBR1* rs10512263: *TC + CC* vs. *TT*	3.80 (1.78–8.12), 0.0006	
		Late toxicity
Acute toxicity ≥ °2		2.62 (1.31–5.21), 0.006
*TGFB1* rs10417924: *GA + AA* vs. *GG*		0.39 (0.18–0.86), 0.019
*TGFB1* rs1800470 (L10P) #: *CC* vs. *TT + TC*		2.70 (1.11–6.53), 0.028
*TGFBR1* rs78471739: *TC + CC* vs. *TT*		0.16 (0.02–1.20), 0.075

Abbreviations: CTCAE = Common Toxicity Criteria assessment for Adverse Events; LENT-SOMA = Late Effects of Normal Tissues − Subjective, Objective, Management criteria with Analytic laboratory and imaging procedures; OR = Odds ratio; CI = confidence interval; PLDA = pelvic lymphatic drainage area; *TGFB1* = transforming growth factor beta-1; *TGFBR1* = transforming growth factor beta receptor-1; ° = grade (in toxicity scoring), *TC = Thymine/Cytosine*; *CC = Cytosine/Cytosine*; *TT = Thymine/Thymine*; *GC = Guanine/Cytosine*; *CC = Cytosine/Cytosine*; *GG = Guanine/Guanine*; *GA = Guanine/Adenine*; *AA = Adenine/Adenine*; *AG = Adenine/Guanine*; rs<number> = unique reference number for single nucleotide polymorphisms (SNPs) according to dbSNP database (www.ncbi.nlm.nih.gov/projects/SNP/, accessed on 19 February 2018). * Fisher exact test. # As in high LD with rs2241713 (r² = 0.97) and rs8108357 (r² = 0.82), rs1800470 (i.e., Leu10Pro), which was much more addressed in literature, was chosen for multivariable analysis in relation to acute or late toxicity.

## Data Availability

All the data is in the Appendix A.

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
