# Peer review of "Identification of Risk Loci for Radiotoxicity in Prostate Cancer by Comprehensive Genotyping of TGFB1 and TGFBR1"

_cancers, 2021, doi:10.3390/cancers13215585_

Round 1

Reviewer 1 Report

Identification of risk loci in response  to radiotherapy is novel and time demanding research which is conducted by the group of Markus Anton Schirmer. The authors have figured out one single nucleotide polymorphism rs10512263 in TGFBR1 By comprehensive genotyping of TGFB1 and TGFBR1. This marker might be considered later in this arena to control the toxicity of radiation therapy in prostate cancer. However, considering that one of the experimental methods has not been well explained and some misunderstandings remained in the present version of this manuscript. Still , some questions are to be discussed.

Major revisions:

  1. The authors need to provide all the used reagents, medium and others with the specifications under creating Material section.
  2. Please describe the details of cell culture and others i.e., number of cells, well plate etc., prior to irradiation of the cells in Assessment of DNA repair capacity.
  3. Could you please mention the duration of the irradiation of the cells in Assessment of DNA repair capacity? On which basis, 3 Gy was considered to apply? Did you try a range of irradiation?
  4. It would be better if you could include the fluorescence images of residual g-H2AX (γH2AX) foci for the TGFBR1 polymorphism rs10512263 and the other two polymorphism compared with the control. You might have a look into the following article

Development of a high-throughput γ-H2AX assay based on imaging flow cytometry | Radiation Oncology | Full Text (biomedcentral.com)

  1. In line 145, in Assessment of DNA repair capacity, the message is not clear “to a portion of these LCLs”. Could you please clarify this sentence?

Minor revisions:

  1. Could you please include the meaning of the abbreviation of TT and TC for all figures and in the text where needed?
  2. To maintain consistency and better understanding I would recommend using the abbreviations TGFβ for transforming growth factor beta and TGFβR1 for transforming growth factor beta receptor 1.
  3. In line 148 please correct “3°h at 4 C” the units.

Reviewer 2 Report

Authors have investigated about the genetic variability in transforming growth factor beta (TGFB) pathway and adverse effects of radiotherapy (toxic side effects or radiotoxicity) on prostate cancer patients. They discovered a high risk for late radiotoxicity in patients who were carrying at least two risk genotypes (polymorphisms) in TGFB1 pathway. Through this specific genotyping of TGFB1 and TGFBR1, authors identified different promising biomarkers for radiotoxicity in prostate cancer.

The topic is original or relevant in the field. Authors described for first time specific genetic variations on TGFB1/ TGFBR1 pathway. Other authors have described previously association between TGFBR1 polymorphisms and cancer risk (breast, ovarian, colorectal…) but not linked to radiotoxicity adverse effect.

A new potential biomarker predictor for late radiotoxicity in prostate cancer patients. It will be useful in order to decide the best treatment for prostate cancer patients.

I do not consider any improvement regarding the described methodology. Methodology is well described; control experiments are included and consistent with the original hypothesis.

The conclusions are consistent with the evidence and arguments presented and address the main question posed. I really think that authors presented through the manuscript and supplemented information a rationale and consistent explanation about the main topic.

The references are appropriate. All the references are associated to the main topic (TGFB) pathway and prostate cancer risk, involvement of this pathway in other types of cancer, polymorphisms association with different cancer pathways, etc.

I think that tables and figures are properly described.

Minor revision:

Page 2. Line 78. Substitute mechanisticactions by mechanistic actions

Page 13 line 332. Include the complete name of IMRT and IGRT abbreviations.

Reviewer 3 Report

  • General comment

The manuscript entitled "Identification of risk loci for radiotoxicity in prostate cancer by comprehensive genotyping of TGFB1 and TGFBR1” aim to investigate the genetic variability of the TGFB1 and TGFBR1 in adverse events related to radiotherapy treatment of prostate cancer.

The manuscript is well written, and the topic discussed is somewhat interesting in the clinical practice. Moreover, many points reported could be a starting point for further research.

Despite those qualities, an improvement in fluency and clarity should be sought in order to improve the readability of the study for interested readers.

Below, the main concerns are reported:

  • Major Corrections

INTRODUCTION

67: As TGFB pathway represent the main topic of this manuscript, it should be better the expand the introduction on this argument, especially in relation to side effects related to radiotherapy.

MATERIAL AND METHODS

99: there is an abundance of self-citations. Are they all necessary? If so, please explain

113: The inclusion of patients which underwent to external radiotherapy and HDR brachytherapy could change the outcomes of acute and late toxicities and this should be reported in the limitations of the study

127: check style as I think this is another paragraph or subparagraph

RESULTS

Table 1: regarding Gleason score, currently a GS < 6 is virtually inexistent. Please explain this finding

204-221: this part of the text should be simplified to improve readability and fluency

DISCUSSION

Limitations should be reported.

  • Minor Corrections

First of all, typos should be checked. In addition, citations in the text should be placed at the end of sentences.

INTRODUCTION

45: I would avoid starting a manuscript with “In men”. A simple rewrite of the sentence could be enough to improve elegancy.

48: citation required

52: Please report some examples and add citations regarding QoL in patients undergoing radiotherapy or radical prostatectomy

59: which are genetic measures?

65: add comma after “factors”

78: Clarity could be improved

RESULTS

194: I would report those four genetic parameters also in the text to improve readability.

Reviewer 4 Report

The authors give an extensive analysis of TGFB1 and TGFBR1 mutations which may be correlated to early and late radiotherapy induced toxicity in prostate cancer patients. The study is well designed and conducted and gives valuable insights into the potential use of these mutations as biomarker for toxicity prediction. I think the manuscript is well written en presents highly relevant research. I have only a few minor comments and questions.

General

-What was the rationale to assess the DNA repair capacity in LCL cells and not PCa cells? And what could findings in the LCL cells potentially mean for PCa toxicity prediction?

Discussion

-Please include some more information on how TGFbeta might influence DSB repair.

-Please include a more elaborate explanation on how findings from this study could be implemented in the clinic. For example, if successful in follow—up studies, what would be the recommendation? Would it be possible to apply this for other cancer types? This information would put the importance of the findings in a broader perspective.

Round 2

Reviewer 3 Report

No further corrections are required